# A MYB-Related Transcription Factor from *Lilium lancifolium* L. (LlMYB3) Is Involved in Anthocyanin Biosynthesis Pathway and Enhances Multiple Abiotic Stress Tolerance in *Arabidopsis thaliana*

**DOI:** 10.3390/ijms20133195

**Published:** 2019-06-29

**Authors:** Yubing Yong, Yue Zhang, Yingmin Lyu

**Affiliations:** Beijing Key Laboratory of Ornamental Germplasm Innovation and Molecular Breeding, National Engineering Research Center for Floriculture, College of Landscape Architecture, Beijing Forestry University, Beijing 100083, China

**Keywords:** MYB, CHS, anthocyanin biosynthesis, abiotic stresses, lily

## Abstract

Most commercial cultivars of lily are sensitive to abiotic stresses. However, tiger lily (*Lilium lancifolium* L.), one of the most widely distributed wild lilies in Asia, has strong abiotic stresses resistance. Thus, it is indispensable to identify stress-responsive candidate genes in tiger lily for the stress resistance improvement of plants. In this study, a MYB related homolog (LlMYB3) from tiger lily was functionally characterized as a positive regulator in plant stress tolerance. LlMYB3 is a nuclear protein with transcriptional activation activity at C-terminus. The expression of *LlMYB3* gene was induced by multiple stress treatments. Several stress-related cis-acting regulatory elements (MYBRS, MYCRS, LTRE and DRE/CRT) were located within the promoter of *LlMYB3*; however, the promoter activity was not induced sufficiently by various stresses treatments. Overexpressing *LlMYB3* in *Arabidopsis thaliana* L. transgenic plants showed ABA hypersensitivity and enhanced tolerance to cold, drought, and salt stresses. Furthermore, we found *LlMYB3* highly co-expressed with *LlCHS2* gene under cold treatment; yeast one-hybrid (Y1H) assays demonstrated LlMYB3 was able to bind to the promoter of *LlCHS2*. These findings suggest that the stress-responsive LlMYB3 may be involved in anthocyanin biosynthesis pathway to regulate stress tolerance of tiger lily.

## 1. Introduction

Environmental stresses, such as drought, high salinity, and extreme temperatures, adversely affect the growth, development, and productivity of plants. To adapt to these environmental stressors, plants have evolved complex signaling cascades to regulate the expression of stress-related genes that can improve stress tolerance either directly or indirectly [1]. Many proteins and genes in the complex signaling networks are regulated by multiple transcription factor (TF) families. As one of the largest TF groups in plants, the MYB family has been proven to be essential for responding to abiotic stresses [2,3].

MYB proteins are characterized by their highly conserved DNA-binding domains [2,4]. According to the number of imperfect repeats of the SANT (for SWI3, ADA2, N-CoR, and TFIIIB) domain (50–53 amino acids) in MYB DNA-binding domains, plant MYB proteins can be mainly sub divided into three subfamilies: MYB-related (one single SANT domain), the R2R3-MYB (two SANT domains) and R1R2R3-MYB (three SANT domains) [5]. Accordingly, research on *MYB* genes has mainly focused on the R2R3-MYB gene family, which has been shown to play important roles in many plant-specific processes including the response to abiotic stress in past decades [6]. In *Arabidopsis thaliana* L., AtMYB14 and AtMYB15 enhance freezing tolerance by regulating *CBF* and its downstream target genes [7,8]; AtMYB20 and AtMYB44 confer salt and drought resistance respectively by downregulating the expression of *PP2Cs* [9,10]; AtMYB2, AtMYB15 and AtMYB96 function in the ABA-mediated drought and salt stress response [11,12,13]. In rice (*Oryza sativa* L.), OsMYB4 was reported to be a positive regulator in transgenic Arabidopsis, tomato (*Solanum lycopersicum* L. cv. Tondino), and apple (*Malus pumila* Mill.) to cold and drought tolerance [14,15,16]; Ectopic expression of *OsMYB2* facilitated salt, cold, dehydration tolerance in rice [17]. In wheat (*Triticum aestivum* L.), overexpression of *TaMYBsm1*, *TaMYB33*, *TaMYB2A* and *TaMYB30-B* have been shown to improve the drought tolerance in Arabidopsis [18,19,20,21]; TaMYB73 can improve salinity stress tolerance in Arabidopsis [22]; TaMYB19 has been found to participate in responses to abiotic stress in transgenic Arabidopsis [23]. However, compared to 2R-MYB genes, there are few reports of functional studies of other MYB subfamilies in abiotic stress response in plants [5]. 

MYB TFs have also been identified to be the major determinant regulators in anthocyanin biosynthesis [24]. Interestingly, a cross-talk exists between abiotic stresses responses and anthocyanin biosynthesis. For instance, low temperature induced and high temperature suppressed anthocyanin biosynthesis in Arabidopsis, which involved the altered regulation of *AtMYB3*, *AtMYB6* and *AtMYBL2* [25,26]. Overexpression of *AtPAP1* or *AtMYB12*, two flavonol synthesis regulators, enhances oxidative and drought tolerance in Arabidopsis [27]. *MaMYB10* in apple, *PcMYB10* in pear (*Pyrus communis* L.) and *BrMYB2-2* in *Brassica rapa* are responsible for the temperature affected anthocyanin accumulation [28,29,30]. Thus, it is pertinent to propose that some MYB proteins may be involved in the correlation between anthocyanin accumulation and abiotic stress tolerance. 

As a wild stress-resistant plant, tiger lily (*Lilium lancifolium* L.) has been shown to have capacity for resisting multiple abiotic stresses [31,32], which could be an ideal model to study stress tolerance mechanisms and signaling regulation of a stress-resistant plant. Based on our RNA-seq data published previously [32], we isolated a MYB-related type gene, *LlMYB3*, from *L. lancifolium*, to further study the role of LlMYB3 in plant stress response. Our work showed that *LlMYB3* was induced by cold, drought, salt and exogenous ABA treatments. Ectopic expression of *LlMYB3* could improve cold, drought and salt tolerance by upregulating the transcription of several stress-related genes in Arabidopsis. Moreover, LlMYB3 TF might be involved in anthocyanin biosynthesis, which can bind to the promoter of *LlCHS2*. These results provide valuable insights into the role of the LlMYB3 in regulating plant stress response.

## 2. Results

### 2.1. Isolation of LlMYB3 and Sequence Analysis

*LlMYB3* gene comprises 1205 bp nucleotides with 462 bp open reading frame, 341 bp 5′ UTR, 402 bp 3′ UTR. It encodes a putative protein of 153 amino acids with a calculated molecular mass of 17.71kD and a pI of 9.54. Amino acid analysis revealed that the LlMYB3 is a MYB-related type protein with one single conserved SANT domain at the N-terminus between 39 and 86 amino acids (Figure 1a). A phylogenetic tree based on the amino acid sequences of some well-studied MYB proteins was constructed [33], which revealed that LlMYB was clustered closely to OsMYB4, OsMYB2 and AtMYB41 (Figure 1b). Furthermore, we BLAST the DNA sequence of *LlMYB3* to the whole-genome sequence of Arabidopsis thaliana using The Arabidopsis Information Resource (TAIR) to locate the chromosomal location. The BLAST result showed that the highest score (bits) significant alignment of *LlMYB3* was *AT5G62320.1* which was located in No.5 chromosome (Appendix A).

### 2.2. Subcellular Localization and Transactivation Assay of LlMYB3

The GFP-LlMYB3 fusion construct and the GFP control in pBI121-GFP vector driven by CaMV35S promoter were transiently expressed in tobacco epidermal cells and visualized under a laser scanning confocal microscope to determine the subcellular localization of LlMYB3. Results showed that the fluorescence signals from GFP alone were widely distributed throughout the cells, whereas, the GFP-LlMYB3 fusion protein fluorescence signal was mainly detected in the nucleus (Figure 1c). Thus, these results demonstrated that LlMYB3 is a nuclear protein. 

To investigate the transcriptional activity of LlMYB3 protein, the entire coding region, N-terminal and C-terminal domain coding sequence were inserted into the pGBDKT7 vector, which contains the GAL4 DNA-binding domain. The transactivation results showed that all transformed yeast cells grew well on SD/-Trp medium (Figure 1d). The yeast strain containing the full-length LlMYB3 (LlMYB3-A) and the C-terminus of LlMYB3 (LlMYB3-C) could grow well on the selection medium SD/-Trp/-His/-Ade, while the cells with the N-terminus of LlMYB3 (LlMYB3-N) and pGBDKT7 empty vector could not grow normally (Figure 1d). Furthermore, the yeast cells that grew well on the SD/-Trp/-His-x-α-gal medium appeared blue in the presence of α-galactosidase, indicating the activation of the reporter gene *Mel1* (Figure 1d). These results indicated that LlMYB3 is a transcriptional activator, and its transactivation domain locates in the C-terminal region.

### 2.3. Expression Patterns of LlMYB3 under Multiple Stresses and ABA

To explore the possible involvement of *LlMYB3* in abiotic stress response, we analyzed the expression patterns of *LlMYB3* in tiger lily plants after treatment with abiotic stresses and ABA. The qRT-PCR analyses revealed that the *LlMYB3* gene has relatively high expression levels in bulb and flower petal, while its expression was low in the leaf and stem (Figure 2a). The expression of *LlMYB3* was significantly and rapidly induced within 1 h after cold, salt and drought treatment, leading to sevenfold to ninefold increase, but only in the salt treatment, the transcript level of *LlMYB3* increased again during 6–24 h (Figure 2c–e). Treatment of tiger lily plants with ABA induced the expression of *LlMYB3*, showing fivefold to sixfold increases at 24 h after treatment (Figure 2b). These data indicate that *LlMYB3* is a stress-responsive MYB-related gene in tiger lily, and its expression is sensitive to cold, drought and salt signaling molecules.

### 2.4. Promoter Analysis of LlMYB3 under Multiple Stresses and ABA

To clarify the mechanism underlying the stress-inducible expressions of *LlMYB3*, the 1374 bp upstream of ATG start codon *LlMYB3* promoter sequence was cloned and used to drive the *GUS* expression in Arabidopsis. Sequences of various putative stress-related cis-acting regulatory elements were identified, including MYB, MYC recognition sites, LTRE, DRE/CRT, TGA and ERE elements (Table 1). 

By histochemical GUS staining, we only observed prominent GUS staining in leaf veins of over 2-week old transgenic *LlMYB3* promoter plants (Figure 3a). There was no obvious difference in GUS staining between stress-treated and non-treated transgenic plants. Thus, the stress inducible activity of the *LlMYB3* promoter was revealed by measuring *GUS* gene expression levels in the transgenic Arabidopsis through qRT-PCR analyses. The result showed that *GUS* gene transcript level could be induced by cold, ABA, salt and drought treatments with a maximal level at 2 and 12 h, respectively (Figure 3b). By GUS enzyme activity assay, however, only an extremely weak fluorescence signal was detected. These results indicated that the activity of *LlMYB3* promoter can be induced by multiple stress treatments, while it is not strong enough to mediate the GUS enzyme activity.

### 2.5. Overexpression of LlMYB3 in Arabidopsis Improves Tolerance to Cold and Drought Stresses

To explore the function of LlMYB3 in providing tolerance to abiotic stress in plants, transgenic Arabidopsis plants overexpressing *LlMYB3* driven by the CaMV35S promoter were generated. Two independent homozygous lines LlMYB3-5 (L5) and LlMYB3-8 (L8) with relatively high expression levels (Appendix A) were selected for the analysis. 

To study the effect of *LlMYB3* overexpression on cold stress, *LlMYB3* transgenic lines and wild-type (WT) plants were grown in equal amounts of potting soil for 4 weeks under normal conditions, and cold stress was applied by being exposed to −4, −6, −8 °C for 12 h. The results showed that all plants grew well under −4 °C treatment as the same as under normal temperature 22 °C (Figure 4a). When the temperature decreased to −6 °C, most of WT plants were dead with a survival rate at approximately 20%, but over half of transgenic plants survived (Figure 4a,b). Furthermore, all WT plants were dead, whereas the survival rate for transgenic plants was observed at 30–35% under −8 °C treatment (Figure 4a,b). In a further experiment, 4-week-old plants were treated at 4 °C for 3 h, and the relative electrolyte leakage and soluble sugars were measured after treatment. As a result, the electrolyte leakage was lower in transgenic plants relative to WT plants (Figure 4c); and transgenic plants produced remarkably higher levels of soluble sugars under a chilling condition compared to WT plants (Figure 4d). 

Similarly, to study drought stress tolerance, after withholding water for 30 days, WT plants showed visible symptoms of drought-induced damage, such as drying, wilting, and even death while some transgenic plants remained green with expanded leaves (Figure 4e). Further analyses showed that after re-watering, few WT plants survived, whereas about 78–82% of transgenic plants continued to grow (Figure 4e,f). Additionally, after being treated with 16.1% PEG6000 (−0.5 MPa) for 3 h, transgenic plants showed lower electrolyte leakage and higher levels of soluble sugars compared to WT plants (Figure 4c,d). The water-loss rates were also slightly lower in transgenic plants (L5) than in WT plants after 3 h treatment (Figure 4f). 

### 2.6. Overexpression of LlMYB3 in Arabidopsis Increases Seed Sensitivity to ABA and Tolerance to NaCl

The salt tolerance and ABA sensitivity of *LlMYB3* transgenic plants was assessed. NaCl significantly inhibited Arabidopsis germination when the seeds were cultivated on MS medium supplemented with 50 mM NaCl (Figure 5a). Only about 30% of the WT seeds germinated in MS medium containing 50 mM NaCl while about 60–65% of transgenic plants seeds were able to germinate (Figure 5a,b). In contrast, except that WT seeds germinated slower in MS medium containing 2 µM ABA, no obvious difference was observed between the germination of WT seeds cultivated on MS medium supplemented with 0 and 2 µM ABA. However, the germination ratio of transgenic plants seeds was remarkably lower than that of WT seeds in MS medium containing 2 µM ABA (Figure 5a,b). Therefore, we suggested that *LlMYB3* transgenic plants are more tolerant to salt stresses and more hypersensitive to ABA than WT plants.

### 2.7. Altered Expression of Stress-Responsive Genes in LlMYB3 Transgenic Plants

The *LlMYB3* transgenic plants exhibited an improved tolerance to freezing, drought and salt stresses. We then measured the expression levels of genes involving stress response in the transgenic plants under normal conditions. Except for *AtCOR47*, transcripts of *AtRD29A*, *AtRD29B*, *AtRD20*, *AtABI5*, *AtGolS1*, *AtLEA14* and *AtAPX2* genes (NCBI accession numbers are shown in Appendix A) accumulated in *LlMYB3* transgenic plants compared to WT plants (Figure 6). The enhanced expression of these genes in transgenic plants might contribute to the stronger stress tolerance, which also implied that LlMYB3 TF may confer stress tolerance through regulating various stress-responsive genes.

### 2.8. LlMYB3 Can Bind to the Promoter of LlCHS2

In our previous study, through the analysis of gene co-expression networks involved in cold resistance of tiger lily, we found that the *LlMYB3* was highly co-expressed with genes involved in anthocyanin biosynthesis pathway, including phenylalanine ammonia-lyase (*PAL*), cinnamic acid 4-hydroxylase (*C4H*), 4-hydroxycinnamoyl-CoA ligase (*4CL*), chalcone synthases (*CHS*) and flavonol synthase (*FLS*) [32]. In this study, the results of qRT-PCR and Pearson’s correlation coefficient (*r*) confirmed that the expression pattern of *LlMYB3* was significantly similar to *LlCHS2*’s (chalcone synthase2) (*r* > 0.8) under continuous cold treatment (Figure 7a and Appendix A). The *LlCHS2* gene information is shown in Appendix A. Thus, we performed the Y1H assay to explore whether there is an interaction between LlMYB3 protein and *LlCHS2* promoter. The 932 bp upstream of ATG start codon *LlCHS2* promoter sequence was cloned, and the fragment (−820 to −553) of the *LlCHS2* promoter containing four MYB binding sites was isolated (Appendix A and Appendix A). The minimal inhibitory concentration of Aureobasidin A (AbA) for bait yeast strains was found to be 200 ng·mL^−1^ (Appendix A). Yeast cells transformed with pGADT7-LlMYB3 and pAbAi-proLlCHS2 grew well on SD/Leu plates with 200 and 250 ng·mL^−1^ AbA (Figure 7b). This showed that LlMYB3 could bind to the promoter of *LlCHS2*; suggesting LlMYB3 is involved in the regulatory pathway of *LlCHS2*.

## 3. Discussion

Considerable studies indicate that plant MYB family members play critical roles in response to abiotic stresses. However, the evidence of improved stress resistance for most *MYB* genes is mainly from model species as Arabidopsis and rice. In this study, a cold stress-responsive gene *LlMYB3* was cloned and characterized from tiger lily (*Lilium lancifolium* L.), to investigate the role of this *MYB* gene response to various abiotic stresses. Sequence analysis shows that LlMYB3 is a MYB-related type protein with one single conserved SANT domain and displays high identity with reported stress responsive MYB member OsMYB4, OsMYB2 and AtMYB41. As predicted that LlMYB3 is a TF, LlMYB3 protein is localized in the nucleus and both the C-terminal and full-length *LlMYB3* have high transactivation ability in yeast. Our previous transcriptome data analysis identified a unigene contig 10,499 coding for LlMYB3, which showed significant changes in expression in tiger lily under cold stress [31,32]. Here, we further confirmed that *LlMYB3* was also up-regulated by drought, salt and ABA treatments. 

Several stress-related cis-elements are present in the promoter of *LlMYB3*. The *LlMYB3* promoter activity is shown to be induced by multiple stress treatments; however, it is not strong enough to mediate the GUS activity significantly. It indicates that the *LlMYB3* promoter is not an effective stress-inducible promoter, and the expression of *LlMYB3* responding to abiotic stresses might mainly be regulated by the upstream regulatory factors. Furthermore, the promoter of *LlMYB3* shows vascular vein specific expression in transgenic Arabidopsis leaves. Usually, an expression pattern of a gene by promoter analysis could reflect its function [34]. The gene expression in leaf veins is usually regulated by the alteration of environments and physiological metabolic signals of this tissue during leaf development and growth [34]. Thus, it is assumed that *LlMYB3* gene might function in vascular tissues in leaves during stress response. 

Compared to R2R3-MYB genes, few studies of the MYB-related genes in abiotic stress response have been reported in plants [23]. For instance, AtMYBC1, a 1R-MYB protein, was reported to be a repressor of freezing tolerance in a CBF independent pathway in Arabidopsis [35]. In rice, MYBS3 was shown to be essential for cold stress tolerance [36]; and overexpression of *OsMYB48-1* enhanced drought and salinity tolerance in rice [23]. In potato, the single MYB domain TF StMYB1R-1 has been shown to involve in drought tolerance by activation of drought-related genes [37]. Overexpression of a single-repeat MYB TF *AmMYB1* from grey mangrove conferred salt tolerance in transgenic tobacco [38]. In the present study, we generated transgenic Arabidopsis plants overexpressing *LlMYB3* gene under the control of the constitutive CaMV35S promoter. Both morphological and physiological evidence strongly demonstrated that transgenic lines had more pronounced tolerances to cold, drought and salt than WT. At gene transcription level, qRT-PCR analysis showed that the expression level of 7 of picked 8 stress-responsive genes, including *AtRD29A*, *AtRD29B*, *AtRD20*, *AtGSTF6*, *AtGolS1*, *AtLEA14* and *AtAPX2* genes were higher in the transgenic plants compared with those of WT. It suggests that these genes might be transcriptionally regulated directly or indirectly by LlMYB3, and overexpressed *LlMYB3* gene may enhance stress tolerance by regulating downstream stress-responsive genes in Arabidopsis. 

Moreover, we showed overexpression of *LlMYB3* resulted in enhanced ABA sensitivity, which was also observed on many MYB TFs from Arabidopsis, such as AtMYB2, AtMYB15, AtMYB96 and AtDIV2 [10,11,12,39]. Given that the expression level and promoter activity of *LlMYB3* can be induced by ABA treatment, *LlMYB3* might be involved in ABA signaling pathway to response to stresses. However, promoter analysis showed that there are no known ABA responsive related cis-acting elements located in the promoter of *LlMYB3*. We thus assume that there are two possible reasons. The first is that novel ABA responsive related cis-elements might exist in the promoter of *LlMYB3*; the second is that the transport of ABA signaling molecules to *LlMYB3* is through a complex signaling network rather than by directly recognizing ABA responsive related cis-elements in *LlMYB3* promoter.

On the other hand, MYB proteins play essential roles by regulating the expression of a large number of anthocyanin biosynthesis genes. For example, in Arabidopsis, the anthocyanin regulators MYB75/PAP1, MYB90/PAP2, MYB113, and MYB114 control the expression of the late biosynthetic genes *DFR* and *LDOX*/*ANS* [2,40,41]; the flavonol regulators MYB12/PFG1, MYB11/PFG2, and MYB111/PFG3 regulate expression of the four early biosynthetic genes *CHS*, *CHI*, *F3H*, *F3’H* and *FLS* [42,43]. More importantly, some anthocyanin biosynthetic genes are even the direct targets of MYB proteins in response to abiotic stresses [3]. MYB1 in carrot can bind to the box-L-like sequences of phenylalanine ammonia-lyase 1 (*PAL1*) promoter specifically and activates *PAL1* under UV-B irradiation [44]. MYB134 in poplar, which is essential for wound and UV-B tolerance, regulates stress-responsive proanthocyanidin biosynthesis by binding to the promoter of proanthocyanidin biosynthetic genes, such as *ANR2* [45]. In rice, OsC1-MYB protein is shown to bind to the MYB responsive elements in the promoters of stress-induced flavonoid pathway genes *OsDFR* and *OsANS* [46]; overexpression of *OsMYB4* in transgenic Arabidopsis increases chilling and freezing tolerance by transactivating *PAL2* and other cold inducible genes [15]. Here, we found LlMYB3 can bind to the MYB binding sites in the promoter of cold-responsive *LlCHS2* gene, suggesting LlMYB3 protein may also function in the correlation between anthocyanin accumulation and cold stress tolerance. 

In conclusion, LlMYB3 is a nucleus-localized transcriptional activator which is regulated by cold, drought and salt stresses and sensitive to ABA. Overexpressing *LlMYB3* in Arabidopsis showed ABA hypersensitivity and enhancing tolerance of transgenic plants to freezing, dehydration and salt conditions by up-regulate many stress-responsive genes. Furthermore, cold-responsive *LlCHS2* is a direct target of LlMYB3 TF in response to abiotic stresses. Therefore, our findings provide a novel MYB-related gene, which plays a positive role in plant stress resistance and might be involved in anthocyanin biosynthesis pathway in response to cold stress. Our future efforts will be focused on investigating the role of upstream regulatory factors in regulating expression and modulating the function of LlMYB3 under various stress conditions.

## 4. Materials and Methods

### 4.1. Plant Materials

The tiger lily seedlings preparation method is described in our previous study [32]. The bulbs of tiger lily were cleaned, disinfected, and then stored at 4 °C; in March, the bulbs were box-cultivated in a greenhouse (116.3° E, 40.0° N) under controlled conditions. The model plant *Arabidopsis thaliana* L. Columbia-0 (Col-0) was selected for the transgenic study of LlMYB3. Arabidopsis plants were grown in 8 cm × 8 cm plastic pots containing a 1:1 mixture of sterile peat soil and vermiculite under controlled conditions (22/16 °C, 16 h light/8 h dark, 65% relative humidity, and 1000lx light intensity). Seeds of *Nicotiana benthamiana* L. were planted and cultured under the same conditions.

### 4.2. Cloning and Sequence Analysis of LlMYB3

The complete sequence cDNA of *LlMYB3* gene was obtained from the transcriptome data of cold-treated tiger lily leave in our laboratory. Primer pairs (Appendix A) were designed to amplify the coding sequence (CDS). The PCR products were cloned into pEASYT1-Blunt vector (TransGen Biotech, Beijing, China). After confirmation by sequencing, plasmid pEASYT1-LlMYB3 was used as a template for all experiments. The homolog genes of *LlMYB3* were searched through BLAST (http://www.ncbi.nlm.nih.gov/BLAST/) database [47]. Multiple sequence alignments were performed using DNAMAN (version 7). Phylogenetic tree analysis was performed using neighbor-joining method in MEGA5 software with 1000 replications [48]. The NCBI accession numbers of genes used in multiple sequence alignments and phylogenetic tree analysis are shown in Appendix A. The theoretical molecular weight and isoelectric point were calculated using ExPASy (http://expasy.org/tools/protparam.html) [49]. 

Genomic DNA was extracted from tiger lily leaves using the DNeasy Plant Mini Kit (Qiagen, Valencia, CA, USA). The promoter of *LlMYB3* gene was isolated using a Genome Walker Kit (Clontech, Mountain View, CA, USA) with nest PCR according to the manufacturer’s instructions. Conserved cis-element motifs of *LlMYB3* promoter were predicted using PLACE (http://www.dna.affrc.go.jp/PLACE/signalscan.html) databases [50].

### 4.3. Abiotic Stresses Treatment and Quantitative Real-Time PCR Analysis

For expression analysis of *LlMYB3* in response to abiotic stress and ABA treatment, 8-week-old tiger lily seedlings were treated with 4 °C, 16.1% PEG6000 (−0.5 MPa), 100 mM NaCl and 100 µM exogenous ABA for 0, 1, 3, 6, 12 and 24 h, respectively. Leaf samples were collected and immediately frozen with liquid nitrogen and stored at −80 °C for RNA isolation.

Total RNA was isolated using an RNAisomate RNA Easyspin isolation system (Aidlab Biotech, Beijing, China). First-strand cDNA synthesis was performed using Prime Script II 1st strand cDNA Synthesis Kit (Takara, Shiga, Japan). The qRT-PCR was performed using a Bio-Rad/CFX Connect™ Real-Time PCR Detection System (Bio-Rad, San Diego, CA, USA) with SYBR^®^ qPCR mix (Takara, Shiga, Japan). Relative mRNA content was calculated using the 2^−ΔΔ^*^C^*^t^ method [51] against the internal reference gene encoding tiger lily tonoplast intrinsic protein 1 (LlTIP1) [31] and Arabidopsis *Atactin* gene (NCBI accession No. NM_112764). The primers used in this study were designed with Primer Premier 5 and are listed in Appendix A. All reactions were performed in three biological replicates. Student’s t-test was performed for all statistical analysis in this study. The heat-map was generated using MeV4.9 and clustered by hierarchical clustering (HCL) with default parameters [52]. Pearson’s correlation coefficient (*r*) to define similarity of expression levels between *LlMYB3* and structural genes involved in anthocyanin biosynthesis pathway.

### 4.4. Subcellular Localization and Transactivation Assay

To determine its subcellular localization, the whole *LlMYB3* coding region without stop codon was amplified and cloned into pBI121-GFP at *Xho*I and *Sal*I by using ClonExpressII One Step Cloning Kits (Vazyme, Piscataway, NJ, USA) to produce LlMYB3-GFP fusion construct driven by CaMV35S promoter. The recombinant constructs and empty GFP vector were transformed into *Agrobacterium tumefaciens* GV3101 and infiltrated separately into tobacco (*N. benthamiana*) epidermal cells. After agro-infiltration for 32–48 h, GFP fluorescence signals were excited at 488nm and detected under Leica TCS SP8 Confocal Laser Scanning Platform (Leica SP8, Leica, Wetzlar, Germany) using a 500–530 nm emission filter. 

The transactivation experiment was carried out according to the manual of Yeast Protocols Handbook (Clontech). The full-length coding region and truncated fragments N-terminus (1–231 bp) and C-terminus (232–462) of *LlMYB3* generated by PCR amplification were fused in frame to the GAL4 DNA binding domain in the vector of pGBKT7 (Invitrogen, Carlsbad, CA, USA). These constructs and negative control pGBKT7 were transformed into yeast strain Y2HGold by using Quick Easy Yeast Transformation Mix (Clontech). The transformed yeast strains were screened on the selective medium plates SD/-Trp, SD/-Trp/-His-Ade and SD/-Trp-His-x-α-gal plates. The transactivation activity was detected according to their growth status and α-galactosidase activity.

### 4.5. Yeast One-Hybrid (Y1H) Assays

Y1H assay was carried out using the Matchmaker™ Gold Yeast One-Hybrid System (Clontech). The *LlCHS2* promoter was amplified by genome walking nested PCR method described previously for *LIMYB3* promoter, and the fragment (−820 to −553) of *LlCHS2* promoter containing four MYB binding sites was isolated and cloned into the pAbAi (bait) vector (shown in Figure 7b and Appendix A). Full-length *LlMYB3* was inserted into pGADT7 (prey) vector yielding plasmid pGADT7-LlMYB3. The bait plasmids were linearized and transformed into the yeast strain Y1HGold. Positive yeast cells were then transformed with pGADT7-LlMYB3 plasmid. The DNA-protein interaction was determined based on the growth ability of the co-transformants on SD/-Leu medium with Aureobasidin A (AbA) according to the manual.

### 4.6. Generation of Transgenic Arabidopsis

The *LlMYB3* open read frame (ORF) was cloned into pBI121 vector under control of a CaMV35S promoter; the *LlMYB3* promoter region was inserted into CaMV35S-GUS vector by replacing the CaMV35S promoter. The recombinant vectors and empty GUS vector were transformed into 5-week-old Arabidopsis ecotype Col-0 plants by using *Agrobacterium tumefaciens* GV3101 and the floral-dip method [53]. Transformed seeds were selected on MS medium containing 50 mg/L kanamycin. T3-generation homozygous lines were selected for gene functional analysis.

### 4.7. Histochemical Staining and Fluorometric GUS Assay

Histochemical staining and fluorometric GUS assay analysis for GUS activity was carried out as described before [54]. To understand the effects of different stresses on GUS expression mediated by the *LlMYB3* promoter, transgenic *LlMYB3* Arabidopsis plants were treated with 4 °C, 16.1% PEG6000 (−0.5 MPa), 100 mM NaCl and 100 µM exogenous ABA for different durations before sampling. The leaves of stress-treated transgenic *LlMYB3* Arabidopsis were incubated in GUS reaction buffer (3 mg/mL X-gluc, 40 mM sodium phosphate pH 7, 10 mM EDTA, 0.1% Triton X-100, 0.5 mM ferricyanatum kalium, 0.5 mM ferrocyanatum kalium and 20% methanol). After overnight incubation at 37 °C, the stained samples were bleached with 70% (*v*/*v*) ethanol to remove chlorophyll. Photos of those stained samples were obtained by a Leica TL3000 Ergo microscope under white light. Leaves of stress-treated transgenic Arabidopsis were also used to exam *GUS* gene expression level by qRT-PCR, and determine GUS enzyme activity and measuring the fluorescence of 4-methylumbelliferone produced by GUS cleavage of 4-methylumbelliferyl-β-d-glucuronide (4-MUG). Protein amount was determined using a Protein Assay kit (Bio-Rad, Hercules, CA, USA) using bovine serum albumin as a standard.

### 4.8. Evaluation of Transgenic Plants Abiotic Stress Tolerance and ABA Sensitivity

The seeds of *LlMYB3* T3-generation homozygous lines and the wild type (WT) were sown on vermiculite soil in pots for freezing and drought treatment. There were 3-week-old seedlings at 4 °C for 3 h, then at −4, −6 or −8 °C, respectively, for 12 h. After that, the plants were kept at 4 °C for 3 h before transferring to a normal condition at 22 °C. For the drought treatment, the water intake of 3-week-old potted Arabidopsis plants in water-saturated substrate was withheld for 30 days, followed by rehydrating the seedlings for 7 days. The survival rates of transgenic and WT seedlings were statistical analyzed. 

For determining the salt tolerance and ABA sensitivity in transgenic plants, Arabidopsis seeds were cultivated on MS medium supplemented with 0 and 2 µM ABA or 50 mM NaCl, respectively, under continuous light at 22 °C in a growth chamber. The germination rate was scored on the 9th day after planting on the plates. 

### 4.9. Measurements of Relative Electrolyte Leakage, Soluble Sugar, and Water Loss Rate

The relative electrolyte leakage, soluble sugar content and water loss rate were evaluated following the method described previously [55,56]. The relative electrolyte leakage was evaluated by determining the relative conductivity of fresh leaves (100 mg) in solution using a conductivity detector. The anthrone-sulfuric acid colorimetry was used for determining the soluble sugar. The water loss rate was calculated related to the initial fresh weight of the leaf samples; the samples were placed on the lab bench (20−22 °C, humidity 45−60%) and weighed at designated time points. All the measurements were performed with ten plants in triplicate.

## Figures and Tables

**Figure 1 ijms-20-03195-f001:**
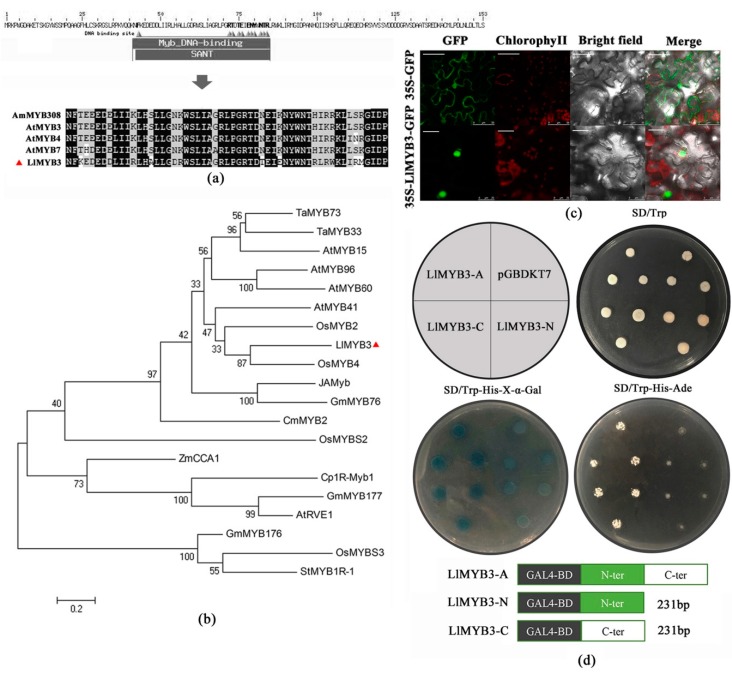
Characterization of tiger lily LlMYB3 protein. (**a**) Alignment of LlMYB3 with *Antirrhinum majus* AmMYB308, *Arabidopsis* AtMYB3, AtMYB4 and AtMYB5. The conserved MYB domain is marked and identical amino acids are shaded in black. (**b**) Phylogenetic tree analysis of LlMYB3 with other known stress-responsive MYB proteins. The LlMYB3 is marked with red triangle. (**c**) GFP and GFP-LlMYB3 fusion proteins were transiently expressed in tobacco leaves under control of the CaMV35S promoter and observed under a laser scanning confocal microscope. Scale bars for 35S-GFP, 50 μm; for 35S-LlMYB3-GFP, 25 μm. (**d**) Full-length protein (LlMYB3-A), N-terminal fragment (LlMYB3-N) and C-terminal fragment (LlMYB3-C) were fused with GAL4 DNA binding domain and expressed in yeast strain Y2HGold. The pGBDKT7 vector was used as a negative control. Transformed yeasts were dripped on the SD/-Trp, SD/-Trp-His-x-gal and SD/-Trp-His-Ade, after being cultured for 3 days in the growth chamber.

**Figure 2 ijms-20-03195-f002:**
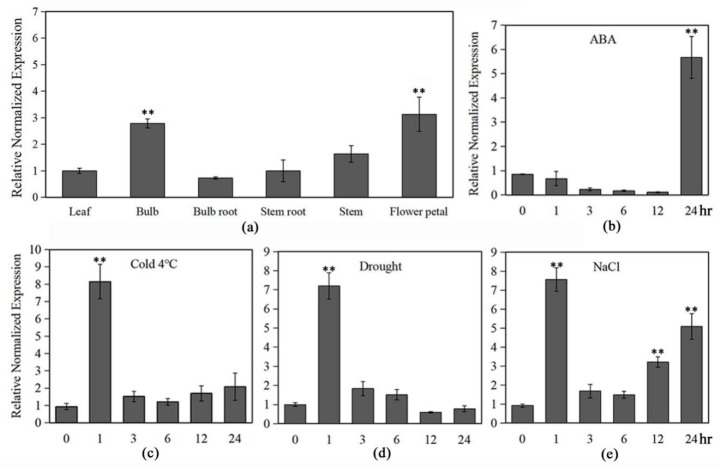
Expression patterns of *LlMYB3* in tiger lily seedlings under different stress treatments. Expression patterns of *LlMYB3* in leaves, bulbs, roots, stems, and flower petal (**a**), and after ABA (**b**), cold (**c**), drought (**d**) and NaCl (**e**) treatments in leaves by qRT-PCR analysis. Transcript levels were normalized to *LlTIP1*. Values are means ± SD of three replicates. Three independent experiments were performed. Asterisks indicate a significant difference (** *p* < 0.01) compared with the corresponding controls.

**Figure 3 ijms-20-03195-f003:**
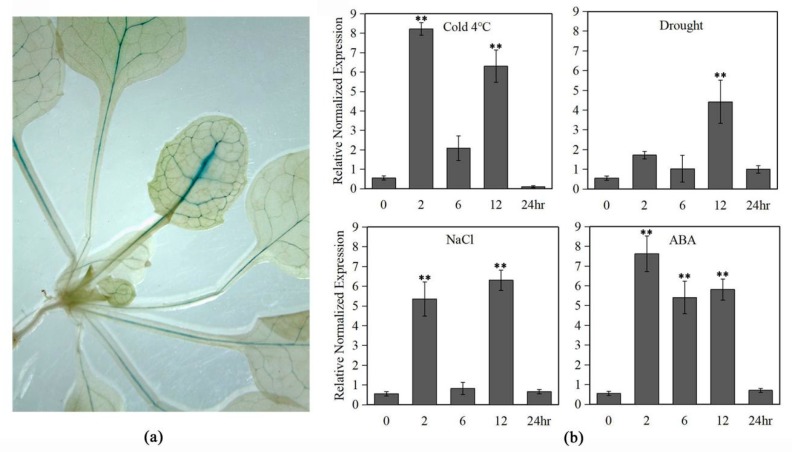
GUS activity of the *LlMYB3* promoter in transgenic Arabidopsis plants. (**a**) Beta-glucuronidase (GUS) expression in untreated transgenic Arabidopsis plants. (**b**) The *GUS* transcript levels in the leaves of the transgenic Arabidopsis under cold (4 °C), drought, salt and ABA treatments. The untreated transformants served as controls. There were 12 transgenic lines acquired. Values are means ±SD of three replicates. Three independent experiments were performed. Asterisks indicate a significant difference (** *p* < 0.01) compared with the corresponding controls.

**Figure 4 ijms-20-03195-f004:**
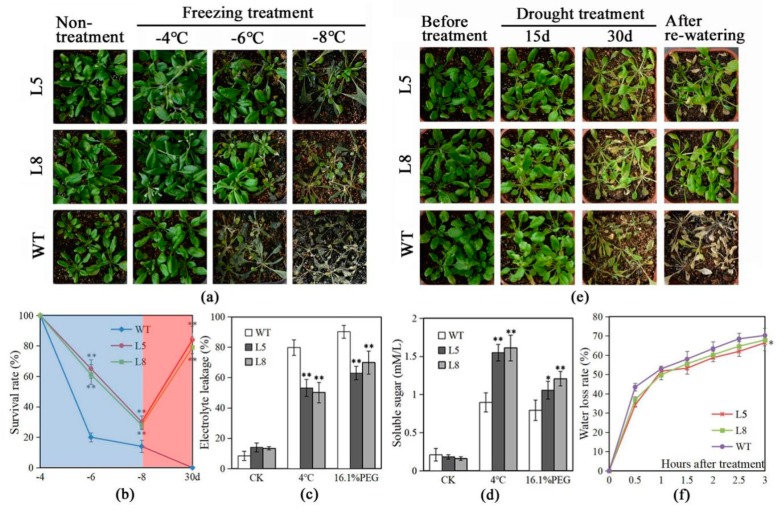
Overexpression of *LlMYB3* in Arabidopsis improves the freezing and drought tolerance. Performance of wild-type (WT) and *LlMYB3* transgenic lines after freezing (**a**) and drought (**e**) treatments. (**b**) Survival rate of plants in (**a**) under freezing temperatures (blue region) and in (**e**) after drought treatment for 30 days (red region). Each data point is the mean of four experiments, and each experiment comprises 30 plants. Relative electrolyte leakage (**c**) and soluble sugar content (**d**) in WT and *LlMYB3* transgenic lines after 4 °C and 16.1% PEG6000 treatments for 3 h. (**f**) Water loss rate of leaves from WT and transgenic Arabidopsis. The data represent the means from three experiments. The bars show the SD. Significant differences between the transgenic and WT lines are indicated as * 0.01 < *p* < 0.05 and ** *p* < 0.01.

**Figure 5 ijms-20-03195-f005:**
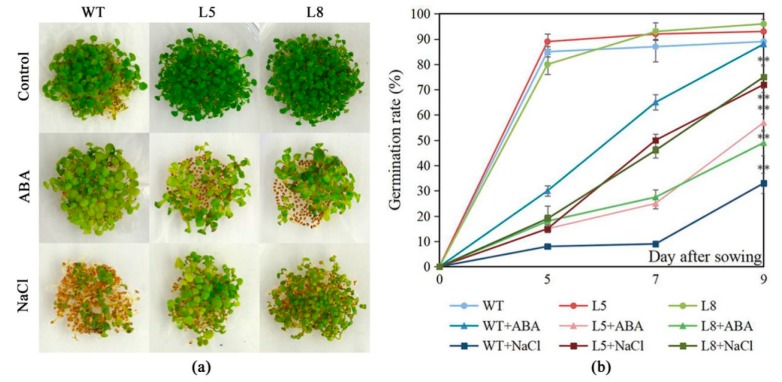
Hypersensitivity and enhancing tolerance of *LlMYB3* transgenic lines to ABA and NaCl. Germination of WT seeds of Col-0 and 35S:LlMYB3 on MS supplemented with 50 mM NaCl and 2 µM ABA after 9 days of incubation at 22 °C (**a**). Germination rate of seeds counted after 5, 7 and 9 days after sowing. The data represent the means from three experiments. The bars show the SD. Significant differences between the transgenic and WT lines are indicated as ** *p* < 0.01.

**Figure 6 ijms-20-03195-f006:**
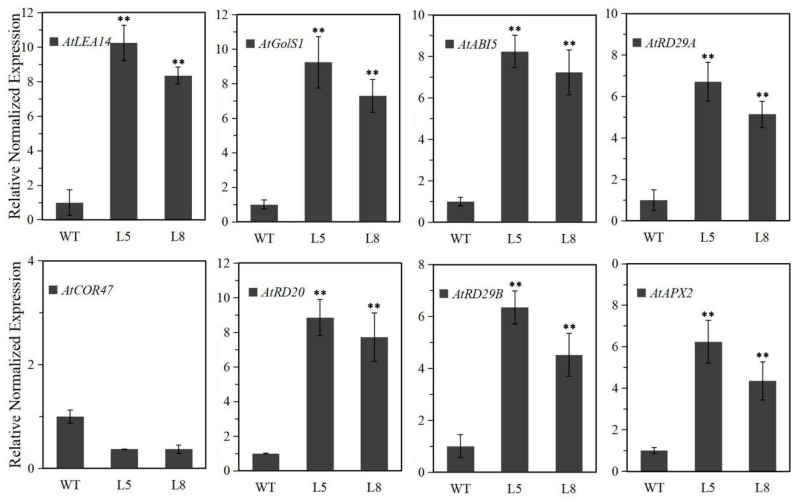
Expression levels of the stress-associated genes under normal condition in WT and *LlMYB3* transgenic plants. Gene-specific primers were used to detect the relative transcript levels of the stress-related genes. Values are means ± SD of three replicates. Three independent experiments were performed. Asterisks indicate a significant difference (** *p* < 0.01) compared with the corresponding controls.

**Figure 7 ijms-20-03195-f007:**
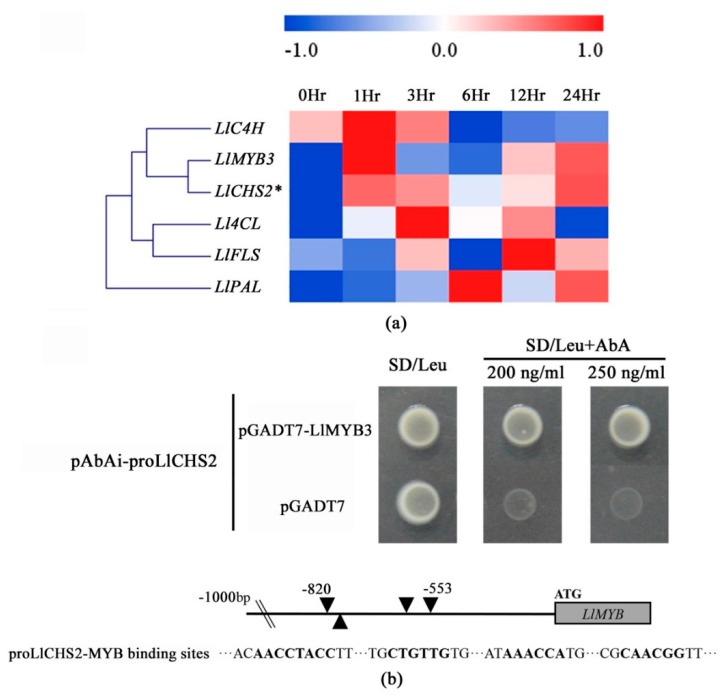
Yeast one-hybrid analysis of LlMYB3 binding to *LlCHS2* promoter. (**a**) Correlation analysis of expression patterns of *LlMYB3* and some structural genes involved in anthocyanin biosynthesis pathway under continuous cold treatment. (**b**) Yeast one-hybrid analysis and schematic representation of MYB binding sites in the *LlCHS2* promoter. Yeast strain Y1HGold was co-transformed with bait (pAbAi-proLlCHS2) and a prey (pGADT7 or pGADT7-LlMYB3) construct. Interaction between bait and prey was determined by cell growth on SD medium lacking Leu in the presence of 200 and 250 ng·mL^−1^ AbA.

**Table 1 ijms-20-03195-t001:** Stress-related cis-acting regulatory elements identified in the promoter region of *LlMYB3*.

Site Name	(Strand) Position	Sequence	Function
ARE	(+)1277	TGGTTT	cis-acting regulatory element essential for the anaerobic induction
CRT/DRE	(+)1323; (−)44	GTCGAC	Core CRT/DRE motif
LTRE	(+)54	ACCGACA	Putative low temperature responsive element
Box I	(+)696	TTTCAAA	light responsive element
MNF1	(−)32	GTGCCCTATA	light responsive element
MBRS	(+)85,039; (−)91	CAACGG (T/A)AACCA	MYB binding site involved in drought-inducibility
MYC	(+)8,438,501,089; (−)5,251,051	CAA(T/C/A)TG CAT(T/G)TG	MYC recognition site involved in cold and drought-inducibility
CGTCA-motif	(+)855; (−)1009	CGTCA	cis-acting regulatory element involved in the MeJA-responsiveness
TGACG-motif	(+)360; (−)484	TGACG	cis-acting regulatory element involved in the MeJA-responsiveness
ERE	(+)695	ATTTCAAA	ethylene-responsive element
TATC-BOX	(+)1274	TATCCCA	cis-acting element involved in gibberellin-responsiveness
TGA-element	(−)490	AACGAC	auxin-responsive element

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
