# Peer review of "A MYB-Related Transcription Factor from Lilium lancifolium L. (LlMYB3) Is Involved in Anthocyanin Biosynthesis Pathway and Enhances Multiple Abiotic Stress Tolerance in Arabidopsis thaliana"

_ijms, 2019, doi:10.3390/ijms20133195_

Reviewer 1 Report

This is a relevant manuscript that could be deemed for publication, but some minor issues should be addressed first.

The authors should report the scientific name of each reported crop when this appeared for the first in the manuscript

Line 308. The authors should briefly describe the method and not only report a reference.

The authors should indicate the pot dimensions and which type of sterile rooting mixture was used, reporting the main chemical characteristics

Line 311 humidity? Light intensity?

Line 409. The authors should briefly describe the method and not only report a reference.

Author Response

Reply to Reviewer 1

Thanks for your comments to this manuscript. We have revised the paper according to all suggestions.

1)       The scientific name of each reported crop has been added when this appeared for the first in the manuscript.

2)       Line 315. We have briefly described the Lilium lancifolium L. seedlings preparation method. The bulbs of tiger lily were cleaned, disinfected, and then stored at 4° C; in March, the bulbs were box-cultivated in a greenhouse (116.3° E, 40.0° N) under controlled conditions.

3)       Line 319. We have indicated the pot dimensions (200px × 200px), the type of used sterile rooting mixture (a 1:1 mixture of peat soil and vermiculite), and the growth conditions (22/16◦C, 16 h light/8 h dark, 65% relative humidity, and 1000lx light intensity).

4)       Line 419. The brief description of the relative electrolyte leakage, soluble sugar content and water loss rate measurements were added.

Reviewer 2 Report

Manuscript entitled “A MYB-Related Transcription Factor from Lilium lancifolium (LlMYB3) is involved in Anthocyanin Biosynthesis Pathway and Enhances Multiple Abiotic Stress Tolerance in Arabidopsis thaliana” by Yong et al describes the isolation and characterization of a Lilium lancifolium gene that codes for a MYB-related protein. The authors have studied this gene by several approaches. The results of subcellular localization in N benthamiana and transactivation assays in yeast suggest that codes for a protein with nuclear localization and transcriptional activation activity.  Also, gene expression studies and promoter sequence analysis shows that could be a gene involved in abiotic stress response in lilly.  As an additional approach to this possibility, the authors perform some transgenic approaches with this gene in Arabidopsis, and determine that the promoter of this gene is also induced under conditions of abiotic stress, as well as with treatment with ABA, and, on the other hand, the overexpression of this gene in Arabidopsis generates plants more tolerant to cold, drought and salt stress and that are hypersensitive to ABA. The authors suggest that this resistance may be due to the activation of the expression of a set of stress response genes. Finally, the authors show that LiMYB3 is able to bind to the promoter of the gene that code for one of the enzymes of the anthocyanin biosynthesis pathway, suggesting again its possible role as transcription factor.

 The proposed approaches and the strategy used for the characterization of the LIMYB3 gene seem both correct for the most part. However, the supplementary material has not been uploaded. This makes it difficult for the reviewer to make a proper review of this work. Also, in my opinion, some aspects of the results and particularly its posterior discussion could benefit from a little improvement to make this work suitable for publication.

Some comments about your paper:

a)Regarding Introduction:

-Lines 35 and 27: Please define MYB and SANT

b)Regarding Results:

-Figure 1 and Lines 86-88, lines 101-104: Please unify the nomenclature of the plasmids used in the transactivational assay. You have different names in figure, figure legend and results text.

-Line 103: please change “selected medium” with “selection medium”

-Lines 105-107: I kindly ask the authors to review the basis of the lac-Z reporter gene assay. This gene codes for a beta galactosidase, which hydrolyses the X-Gal reagent. One of the products of this hydrolysis gives rise to the blue color molecule. Therefore, a more correct sentence would be "... appeared blue in the presence of X-gal, indicating the activation of the reporter gene lacZ."

- Line 108: please change “locates in the C-terminal” with “locates in the C-terminal region”

-Lines 112-113 and Figure 2a.  The sentence “QRT‐PCR analyses revealed that the LlMYB3 gene has relative high expression levels in bulb and flower petal, while its expression was low in the leaf and stem (Figure 2a)” should be backed with statistical analysis, that is absent in Figure 2a.

-Line 1. Table 1 “function column” should include references that describe these functions

-Lines 140-142: Please rephrase the sentence. The promoter of a gene is not capable of mediating an enzymatic activity.

-Lines 143-145. Figure 3 legend should include information about how many transgenic lines were assayed, as well as the number of independent experiments performed.

-Figure 4. Statistical analysis is absent in figure 4b and 4f. This analysis is necessary to back up the statement made in lines 178-179 “The water‐loss rates were also slightly lower in transgenic plants than in WT plants (Figure 4f)”

-Line 195: again, Figure 5b lacks from statistical analysis that is necessary to back up the description of these results in lines 182-191.

-Line 215: Yong et al 2018 reference should appear in number format.

-Line 217. LICHS2 definition (chalcone synthase 2) should appear here and not in line 366

-Line 226: Figure 7. Plasmid nomenclature should be the same in figure, figure legend and text.

c)Regarding Discussion:

-Lines 247-250: I must insist to the authors that the promoter of a gene is not able of mediating an enzymatic activity. On the other hand, the statement of the authors "LiMYB3 is not an effective stress-inducible promoter" should be modified, since it is in contradiction with the results observed in Figure 3b.

-Lines 272-279: The authors should reconsider this part of the discussion. The increased ABA sensitivity of the LIMYB3-overexpressing plants is not an observation that has to be necessarily related to the presence of ABA-responsive elements in the promoter of this gene. 

In the absence of other posttranscriptional or posttranslational regulatory mechanisms, we can assume that a plant that overexpresses a gene has higher levels of the protein for which it encodes. The hypersensitivity phenotype to ABA in plants with higher levels of LIMYB3 suggests that this protein participates in some way in the ABA signal transduction pathway, and that it must be performing some type of activating function, but these results from overexpressor plants cannot prove anything regarding the regulation of the gene that codes for the LIMYB3 protein. In any case, the results of the ABA treatment experiments (Figures 2b and 3b) are those that suggest the involvement of this phytohormone in the control of LIMYB3 expression, despite the absence of ABA-responsive elements in its promoter.

-Line 281: “biosynthesis genes” is a term more accurate than  “biosynthetic genes” 

d)Regarding material and methods

-Line 319: BLAST software should be referenced

-Line 321: MEGA 5 software should  be referenced

-Line 324: Expasy software should  be referenced

-Line 329: PLACE database should be referenced

-Line 339: deltadeltaCt method should be referenced

-MEV 4.9 software should be referenced

-Line 378: Floral dip transformation method should be referenced.

-Statistical methodology should be detailed.

Author Response

Reply to Reviewer 2

Thanks for your comments to this manuscript. We have revised the paper according to the suggestions.

a) The supplementary materials have been uploaded.

b) Regarding Introduction:

Line 37 to 41. We have added the definition of MYB protein and SANT domain.

c) Regarding Results:

1)       Figure 1 and Lines 90-92, lines 105-107. The nomenclature of the plasmids used in the transactivational assay has been unified.

2)       Line 106: The “selected medium” has been changed with “selection medium”

3)       Lines 105-107. We have reviewed the function of x-α-gal. The SD/-Trp/-His-x-α-gal medium appeared blue in the presence of α-galactosidase, indicating the activation of the reporter gene Mel1 instead of LacZ.

4)       Line 111: “locates in the C-terminal” has been changed with “locates in the C-terminal region”

5)       Statistical analysis has been added in Figure 2a to back up the results.

6)       In Table 1, the cis-acting regulatory elements functions described in the “function column” were referenced to the PLACE databases which have been referenced in the Materials and Methods.

7)       Information about how many transgenic lines was assayed, and the number of independent experiments performed has been added in Figure 3 legend.

8)       Statistical analysis has been added in Figure 4b, 4f and Figure 5b to back up the results.

9)       Line 221. Yong et al 2018 reference has been changed in number format.

10)    Line 223. LICHS2’s definition was added.

11)    Line 237. The nomenclature of the plasmids used in the Yeast one-hybrid analysis has been unified.

d) Regarding Discussion:

12)    Lines 279-288. We have revised this part of the discussion according to the review.

13)    Line 290. “biosynthetic genes” has been replaced as “biosynthesis genes”.

14)    Line 355. Statistical methodology has been detailed as “Student’s t-test was performed for all statistical analysis in this study.”

e) Regarding material and methods

15)    BLAST, MEGA 5, ExPASy, PLACE and MEV 4.9 software or database, and 2-△△Ct method and Floral dip transformation method have been referenced.

As for the comments that the sentence “These results indicated the activity of LlMYB3 promoter can be induced by multiple stress treatments, while it is not strong enough to mediate the GUS enzyme activity.” should be rephrase as “The promoter of a gene is not capable of mediating an enzymatic activity.”, and the statement “LlMYB3 is not an effective stress-inducible promoter” is in contradiction with the results observed in Figure 3b, we suppose they are inappropriate. Many inducible promoters have been identified by mediating the reporter GUS enzyme activity under certain conditions [1-3], thus, in our opinion, a inducible promoter is capable of mediating an enzymatic activity when its promoter activity it is induced sufficiently.

[1] Malnoy, M.; Reynoird, J.P.; Borejsza-Wysocka, E.E.; Aldwinckle, H.S. Activation of the pathogen-inducible Gst1 promoter of potato after elicitation by Venturia inaequalis and Erwinia amylovora in transgenic apple (Malus × domestica). Transgenic Res 2006, 15, 83–93, doi:10. 1007/s11248-005-2943-7.

[2] Kim, K.-Y.; Kwon, S.-Y.; Lee, H.-S.; Hur, Y.; Bang, J.-W.; Kwak, S.-S. A novel oxidative stress-inducible peroxidase promoter from sweet potato: molecular cloning and characterization in transgenic tobacco plants and cultured cells. Plant Mol Biol 2003, 51, 831-838, doi:10.1023/A:1023045218815.

[3] Tao, Y.; Wang, F.; Jia, D.; Li, J.; Zhang, Y.; Jia, C.; Wang, D.; Pan, H. Cloning and Functional Analysis of the Promoter of a Stress-inducible Gene (ZmRXO1) in Maize. Plant Molecular Biology Reporter 2015, 33, 200-208, doi:10.1007/s11105-014-0741-1.

 Thank you very much 

Best regards

sincerly yours

Yingmin Lyu

Round  2

Reviewer 2 Report

Dear authors

The revised version of the manuscript has improved with the changes made, but there are still some issues that should be addressed:

a)Supplemental material is again absent. Because of this lack, I am afraid that I should keep my overall recommendation of "major revision" I would like to be able to examine these figures in detail before my final decision about the work.

b)Regarding Figures: Figures 1, 2, 4, 5 and 7 appear duplicated (previous version and new version of each figure). Please correct this issue.

c)Lines 186-187: The time point where there is  a statistical difference in water loss rate between WT and LIMYB3 overexpressing lines should be detailed.

d)Lines 371-372. Details of the generation of the pGBDKT7 constructs containing truncated versions of the gene (N-terminal or C-terminal region) should be added. Also, an indication of the length and location of these regions in the sequence of Figure 1 could be useful for the reader.

e)Line 226: please correct the sentence “The LlCHS2 gene information was shown in Supplementary Figure 1.” with “The LlCHS2 gene information is shown in Supplementary Figure 1.”

f)Line 280: Please  correct “by regulating downstream stress‐responsive genes Arabidopsis” with “by regulating downstream stress‐responsive genes in Arabidopsis”

Author Response

Reply to Reviewer 2

Dear Rewiewer:

Thanks for your comments to this manuscript. We have revised the paper according to the suggestions.

a) The supplementary materials have been uploaded.

b) The issue that Figures 1, 2, 4, 5 and 7 appear duplicated (previous version and new version of each figure) has been corrected.

c) Lines 184-185: The time point where there is a statistical difference in water loss rate between WT and LIMYB3 overexpressing lines has been detailed as “The water-loss rates were also slightly lower in transgenic plants (L5) than in WT plants after 3 h treatment”.

d) Lines 371-372. Details of the generation of the pGBDKT7 constructs containing truncated fragments of LlMYB3 N-terminus (1–231bp) and C-terminus (232–462) have been added. An indication of the length and location of these regions in the sequence also has been added in Figure 1.

e) Line 226: The sentence “The LlCHS2 gene information was shown in Supplementary Figure 1.” has been corrected with “The LlCHS2 gene information is shown in Supplementary Figure 1.”

f) Line 280: The sentence “by regulating downstream stress‐responsive genes Arabidopsis” has been corrected with “by regulating downstream stress‐responsive genes in Arabidopsis”

Best Regards

Sincerely yours

Yingmin Lyu

Round  3

Reviewer 2 Report

Dear authors:

After reviewing the revised manuscripts and supplementary material, I have seen that there are still a number of issues that should be addressed:

a)Figure 1 appears again duplicated (previous version and new version)

b) Lines 248, 435 and 436: replace “LINAC2” with“LIMYB3” (in italics)

c)Please cite supplementary tables and figures in the main text with the appropriate format :

Line 223: please correct “Supplement Table 4” with “Supplementary Table S4”

Line 224: please correct “Supplementary Figure 1” with “Supplementary Figure S1”

d)Regarding supplementary Figure S1: the information about LICHS2 gene is incomplete and is not very useful to the reader, because the coding sequence is not used in any experiment performed. It should be more useful to include the complete sequence of the gene, including the promoter region (and an indication of location of the promoter region cloned and used in the yeast one hybrid assay).

e)My main concern is about supplementary figure S2:

-semi‐quantitative RT‐PCR conditions and primers used should be detailed in material & methods.

-The figure should have a more descriptive figure legend, as in the rest of figures of the paper. The authors should remember that a figure legend should  have enough information to allow to the reader to understand and interpret the figure without having to resort to the main text.

- The figure does not include a negative control (untransformed Arabidopsis plants)

- Since the majority of the gel lanes have the LIMYB3 amplification band with similar intensities, as a reader it is very difficult to me to understand why lines 5 and 8 were chosen. The authors should justify this election with some kind of quantification of the band intensity. 

As a suggestion, this figure could be replaced with an equivalent qRT-PCR figure.

f)Figure S3 legend should be more informative about the experiment performed (as suggested with figure S2). 

g)Supplementary table S3 is not cited in the main text 

h)Supplementary table S4 legend should include the statistic criteria considered to mark with an asterisk.

Some typos detected:

Lines 115, 349: replace "QRT‐PCR” with “qRT-PCR”

Line 119: “replace “(Figure 2c, d, e).Treatment” with “(Figure 2c, d, e). Treatment” (a space is absent)

Line 168: replace “figures 4d” with” Figure 4d”

Line 174: replace “16.1%PEG” with “16.1% PEG6000”

Author Response

Reply to Reviewer 2

Thanks for your comments to this manuscript. We have revised the paper according to all the suggestions.

a) The issue that Figures 1 appear duplicated (previous version and new version) has been corrected.

b) Lines 248, 435 and 436: “LINAC2” has been replaced with“LIMYB3” (in italics).

c) Supplementary tables and figures have been cited in the main text with the suggested format :

Line 223: “Supplement Table 4” has been correct with “Supplementary Table S4”.

Line 224: “Supplementary Figure 1” has been with “Supplementary Figure S2”.

d) Regarding Supplementary Figure S2 (previous version Supplementary Figure S1): the promoter region including an indication of location of the promoter region cloned and used in the yeast one hybrid assay was exhibited in Supplementary Figure S2. However, we are afraid that we cannot display the complete sequence of LlCHS2 gene. The sequence of LlCHS2 gene (from -223 to +196 shown in Supplementary Figure S2) was obtained from our RNA-seq data published previously. Based on this sequence, we designed GSP1 and GSP2 (shown in Supplementary Table S1 and Figure S2) primers to clone the promoter region of LlCHS2. Thus, the complete sequence of LlCHS2 was not isolated in this study.

e) Regarding Supplementary Figure S1 (previous version Supplementary Figure S2): we have replaced this figure with an equivalent qRT-PCR figure. The cDNA used in semi‐quantitative RT‐PCR, and the primers used in analyzing “expression patterns of LlMYB3 in tiger lily seedlings under different stress treatments” were utilized in this qRT-PCR analysis. Wild type Arabidopsis is served as negative control. A more descriptive figure legend of Supplementary Figure S1was also added.

f) A more descriptive figure legend of Supplementary Figure S3 was added.

g) Supplementary table S3 has been cited in the main text (Line 228).

h) The statistic criteria considered to mark with an asterisk has been added in Supplementary table S4 legend.

i) Lines 115, 349: we have replaced "QRT‐PCR” with “qRT-PCR”.

Line 119: we have replaced “(Figure 2c, d, e).Treatment” with “(Figure 2c, d, e). Treatment” (a space is absent).

Line 168: we have replaced “figures 4d” with” Figure 4d”.

Line 174: we have replaced “16.1%PEG” with “16.1% PEG6000”.

Best regards

Yingmin Lyu 

Round  4

Reviewer 2 Report

Dear authors

I am happy to see that the main issues pointed previously have been addressed. Still, some minor issues remain:

a)Be advised that in this currrent version, supplementary figure S3 and tables S1, S2, S3, S4 have not been uploaded.

b)Regarding LlCHS2 Promoter sequence used in Y1H assay:

this sequence was isolated by genome walking nested PCR using GSP1 and GSP2 primers?

This information should be included in line 378 sentence “LlCHS2 promoter was amplified by PCR” (perhaps, with something like "LlCHS2 promoter was isolated with the genome walking nested PCR method described previously for LIMYB3 promoter”)

my main issue this time:   in Material & methods, line 378, the authors indicate that “A fragment (‐371 to ‐643) of the LlCHS2 promoter was amplified by PCR, and cloned into the pAbAi (bait) vector” (a region of 272 bp), but in lines 226-227  and supplementary  figure S2 and its legend in lines 439-440 this region is indicated as “the fragment (−820 to −553) of the LlCHS2 promoter containing four MYB binding sites was isolated” (a region of 267 bp and different location). Please correct or clarify.

Also, a reference to Supplementary figure S2 could also be included in line 379, like “(shown in Figure 7b and supplementary Figure S2)”.

c)lines 437-438: Please change “The line 5 and line 8 showed relative high expression levels of LlMYB3 transcripts were selected for further study” with “The lines 5 and 8, which showed relative high expression levels of LlMYB3 transcripts, were selected for further study”

d)some typos are still present: 

Line 223: replace “Supplement Table 4” with “Supplementary Table S4”.

Line 168: replace “Figures 4d” with” Figure 4d”

Line 174: replace “16.1%PEG” with “16.1% PEG6000”

Author Response

Reply to Reviewer 2

Thanks for your patient and attentive checks to this manuscript. We are sorry for our carelessness. We have revised the paper according to all the suggestions.

a) Supplementary Figure S3 and Tables S1, S2, S3, S4 have been uploaded.

b) Regarding LlCHS2 Promoter sequence used in Y1H assay:

Line 378 the sentence “A fragment (‐371 to ‐643) of the LlCHS2 promoter was amplified by PCR, and cloned into the pAbAi (bait) vector” has been changed with “LlCHS2 promoter was isolated with the genome walking nested PCR method described previously for LIMYB3 promoter, and the fragment (‐820 to ‐553) of the LlCHS2 promoter containing four MYB binding sites was isolated and cloned into the pAbAi (bait) vector”.

The reference to Supplementary figure S2 has been included in line 379, as “(shown in Figure 7b and Supplementary Figure S2)”.

c) Lines 439-440: the sentence “The line 5 and line 8 showed relative high expression levels of LlMYB3 transcripts were selected for further study” has changed with “The lines 5 and 8, which showed relative high expression levels of LlMYB3 transcripts, were selected for further study”.

d) Line 223: “Supplement Table 4” has been replaced with “Supplementary Table S4”.

Line 168: “Figures 4d” has been replaced with “Figure 4d”.

Line 174: “16.1%PEG” has been replaced with “16.1% PEG6000”.

Best regards to you

Sincerely yours

Yingmin Lyu